# The relationships between box turtle gut microbiomes and personality

Kaija Harlow[1]*, Elizabeth K. Service[1], Jace E. Geiger[1], Bradley E. Carlson[2], Steven J.A. Kimble[1]

1 Department of Biological Sciences, Towson University, Baltimore, Maryland, United States of America,
2 Department of Biology, Wabash College, Crawfordsville, Indiana, United States of America

☯ These authors contributed equally to this work.
* kaijaharlow@gmail.com

## Abstract

Though well understood in model animals, the microbiome-brain axis has only recently been found to be important in wildlife health. Host behavior is an important component of this axis, and variation in individual animal personalities can shape the dynamics, composition, and evolutionary trajectory of an ecosystem. Relative boldness or shyness is a commonly used personality metric across vertebrate animals, and the composition of an organism's microbiome may be important in shaping this trait. Turtles have been poorly characterized in terms of behavior and microbiome, though they are the most threatened vertebrate group on Earth and therefore warrant more study. To address this gap, we compared the gut microbiota of bold and shy Eastern Box Turtles to investigate any correlation between microbial composition and personality type. Free-ranging Eastern Box Turtles were captured and assayed for their personality, and cloacal, skin, and oral swab samples were collected. Microbial DNA from swabs was amplified and sequenced. Female turtles had significantly higher alpha microbial diversities than male turtles and shy turtles had a significantly higher microbial diversity than bold turtles. Four significantly overabundant bacteria were found (two in bold turtles and two in shy turtles). The steroid biosynthesis pathway was significantly overrepresented in bold turtles, while the bile acid biosynthesis pathway and purine degradation to urea modules were significantly overrepresented in shy turtles. Firmicutes, Actinobacteria, and Proteobacteria are implicated in the classification of Eastern Box Turtle personality. While our conclusions should be considered in the context of limited sample sizes, our data suggest prominent metabolic pathways, modules, and bacteria that are implicated in different Eastern Box Turtle personality types, suggesting a possible microbial influence on the personality of this well-known but threatened species.

**Data availability statement:** The data underlying the results presented in the study are available at NCBI's Sequence Read Archive via the accession number PRJNA1294286.

**Funding:** Sequencing was funded by a Towson University Faculty Development and Research Committee grant to SK (https://www.towson.edu/). Computational resources were provided by a National Science Foundation ACCESS grant BIO230011 to SK (https://www.nsf.gov/). No funders played any role in the study design, data collection and analysis, decision to publish, or preparation of the manuscript.

**Competing interests:** The authors have declared that no competing interests exist.

## Introduction

Animal personalities can affect the fitness of individuals and the evolutionary trajectories of populations [1, 2]. There are a variety of vertebrate animals that are considered to have "personalities," which can be defined as repeatable variation in behavioral propensities among conspecifics [3–5]. The evolutionary drivers of individual variation in personality are poorly understood but various theories have been proposed, attributing these differences to negative frequency-dependent selection, individual niche specialization, or covariance with stable differences in state [6, 7]. A particular aspect of an individual's state that could drive consistent behavioral differences is physiology, including differences in metabolic rates, hormone levels, and microbiome [8]. Ultimately, a better understanding of the role of personality variation in an ecosystem can aid population management and conservation efforts, as these individual differences can shape abiotic and biotic interactions and affect fitness levels [9, 10].

One commonly used personality measure is relative boldness or shyness of an individual, as it can be easily measured in a variety of different vertebrate taxa (fish, reptiles, birds, and mammals), corresponds with highly repeatable behaviors, is variable among individuals, and can significantly impact ecological processes [4,11]. With the tendency to participate in riskier behavior, bolder individuals are theoretically expected to experience lower survival rates but increased growth or fecundity due to increased resource acquisition [12], though support for this pattern is mixed [13]. Furthermore, bolder personality is often correlated with higher levels of exploration, aggression, and activity [13], which constitute other dimensions of animal personality [4]. A population comprised of individuals with a mix of phenotypes (e.g., bold and shy) may improve its stability [14] and may be a requirement for the population to evolve [15].

Intrapopulation variation in behavior is partly driven by genetic factors, [16] but can also be driven by environmental factors, including the microbiome [17]. The microbiome can be defined as a distinctive community of microbes in addition to the interactions and molecules they share [18], whereas microbiota refers to the collection of organisms that live on or within a living being [19]. Previous research on humans and model species suggests the microbiome has a notable effect on the function of an organism. Specifically, this research has revealed that the gut microbiome can influence behavior, metabolism, and digestion of the host [20]. Conversely, the microbiome is influenced by an array of host-associated factors like habitat, diet, and social interactions [21]. As a result, the microbiome is believed to have a significant effect on the central nervous system (with the interaction between the two being termed the microbiome-gut-brain axis) and thus the behavior and cognition of living organisms [19,22].

In terms of behavior, the gut microbiome can influence certain processes using direct and indirect mechanisms. Directly, gut microbiota produce short-chain fatty acid metabolites that communicate chemically with the nervous system [23, 24]. These metabolites are formed through metabolic pathways, or successions of coupled enzymatic reactions [25]. Many metabolic "steps", or modules, that comprise a

metabolic pathway are independently distributed within different microbes; thus, they are dependent on one another to complete a metabolic pathway [26]. As such, microbial composition is important in predicting the metabolic processes able to be carried out by a host organism. For example, imbalances in the gut microbiome have been found to impact response to stress by disrupting neurotransmitter release and reuptake [27]. As stress is an important determinant in animal behavior, the microbial composition of an organism may be important in shaping behavioral tendencies [28].

Various studies have unearthed relationships between the boldness of individuals and their microbiota. For example, in free-ranging Tibetan macaques (*Macaca thibetana*), significantly lower among-individual microbiota variation was found in bold individuals than shy individuals [24]. In addition, shy macaques have enhanced levels of *Ruminococcus* spp. and *Oscillospiraceae* spp., both of which have been positively associated with negative emotions and/or depression [24]. In similar studies with Mongolian gerbils (*Meriones unguiculatus)*, bold individuals receiving transplanted feces of other bold individuals increased their levels of boldness, and bold individuals were found to have higher amounts of the bacteria *Odoribacter* spp. and *Blautia* spp [23]. These taxa are known proponents of butyrate production, which has been found to improve glucose tolerance, insulin sensitivity, and other components of host metabolism. Their presence, therefore, likely drives the high metabolic rates and activity levels that are characteristic of bold personalities [23,29].

Although the gut microbiome has been researched extensively in recent years, most of this work has centered around humans and model mammalian organisms; there remains a lack of understanding concerning the role of the microbiome in mediating the behaviors of non-model, non-mammalian vertebrates [20]. Nonetheless, recent technologies and methods are allowing for the sequencing of large quantities of DNA from non-model microbiomes. As a result, scientific knowledge is expanding to include more detailed information about the ecological relationships of non-model organisms and their microbes. This exploration of non-model microbiomes can also help with conservation of threatened taxa [30].

Turtles comprise the most threatened major vertebrate group on the planet, facing both substantial environmental and anthropogenic barriers such as habitat loss and modification, pollution, and human recreation [31]. Eastern Box Turtles (*Terrapene carolina carolina*) are a threatened yet widespread reptilian species primarily found in the eastern United States of America [32]. Previous research has shown that Eastern Box Turtles display consistent variation in individual behavioral responses to humans, with bolder turtles emerging from the shell more quickly after standardized handling and confinement [32, 33]. It has been suggested that Eastern Box Turtles maintain individuality in their level of boldness from a juvenile age, keeping these behavioral differences for multiple years [34]. Turtles that emerge from their shell sooner are thought to be at a greater risk for predation, but may capitalize on opportunities to acquire resources, such as food or mates [32,34]. As an endangered, non-model, long-lived organism, characterization of the Eastern Box Turtle personality and potential ecological stressors can provide novel and important information about the role of the gut microbiome in a wild context, while additionally aiding in the development of effective methods for protecting and maintaining threatened populations.

Here, we compare gut microbiota composition between bold and shy Eastern Box Turtles using field-collected cloacal, oral, and skin samples. Our primary objective was to gain insight into the relationship between the microbiome and Eastern Box Turtle behavior by examining differences between the microbiota of bold and shy Eastern Box Turtles, as there is a lack of studies on microbiome-behavior relationships in reptiles. We hypothesized that the microbiota composition of bold and shy turtles would be significantly different. Specifically, we predicted that the microbiota would have significantly different beta diversities (community differences in bold and shy turtles) as has been demonstrated in macaques [24] (but not zebrafish [35]). In addition, we predicted that alpha diversity (microbial richness) would be significantly higher in bold turtles, due to the tendency of bold individuals to be more exploratory and thus encounter a larger portion of their ecosystem [36]. Our second objective was to investigate the relationship of the found microbiota to metabolic pathways and personality traits. We hypothesized that bold turtles would have significantly higher predicted metabolic pathway activity that promotes increased metabolism and aggression, while shy turtles would have significantly higher predicted metabolic pathways that promote chemical messengers associated with depression.

## Methods

### Ethics statement

This study was reviewed and approved by the Wabash College Institutional Animal Care and Use Committee and by the Indiana Department of Natural Resources (Scientific Purposes License #3280). Procedures were consistent with standard practices in turtle handling and veterinary assessment and the duration of handling was minimized to reduce stress [37]. To limit potential discomfort, turtles were swabbed using the procedure from [38] No adverse effects on study subjects were observed in the course of this work.

### Study subjects and sites

The Eastern Box Turtle, a relatively small turtle species that is primarily terrestrial, inhabits wooded environments through-out the Eastern United States [39]. Although they are widespread, Eastern Box Turtles are threatened from loss of habitat, overexploitation, road mortality, and disease [40, 41]. Eastern Box Turtles were used to investigate behavior in this study due to their need for management (to sustain healthy populations), their longevity (which makes them ideal for longitu-dinal behavioral studies), and the dire need for more data on their behavior and microbiome as reptiles and non-model organisms [42]. Eastern Box Turtles were collected from three riparian sites in central Indiana: one from Weiler-Leopold Nature Preserve in Warren County along the Wabash River, three from an upland forest property along Sugar Creek in Montgomery County, and eight from Allee Memorial Woods (AMW), a nature preserve in Parke County, Indiana that spans 72 ha and is also along Sugar Creek. All are comprised primarily of old-growth forest, with some secondary growth. AMW has been used to monitor box turtle population dynamics since 1958 [43]. As a result, seven turtles encountered in this study had been previously captured and two out of the seven had been previously tracked using radiotelemetry to collect behavior-related data [32].

### Field data collection

Eastern Box Turtles were captured from May to June of 2021. All box turtles were found using visual encounter sur-veys. Turtles from AMW had their shell equipped with a radiotransmitter and thus were relocated at a later timeframe for subsequent data collection. Five turtles that had not been equipped with a radiotransmitter were handled and marked in earlier research studies. Head emergence behavioral assays were conducted first, followed by swab collection, and finally measurements were taken (weight, carapace length and width, plastron temperature). This order was followed to prevent potential contamination from measuring tools prior to swabbing, and to prevent handling from influencing prior measurements.

### Behavioral assays

Upon identification and localization of an Eastern Box Turtle, head emergence behavioral assays were performed as described in [32]. Each turtle was immediately placed inside a zippered sterile black opaque bag (16 x 30 cm) with ster-ile gloves. The bag was then placed on the ground for three minutes. This was intended to act as a moderate and novel stressor that was consistent across individuals. After three minutes, the turtle was removed from the bag and placed back in the environment; the researcher then walked approximately ten meters away and remained silent and unmoving. From this distance, the time from placement on the ground to head emergence (head emergence latency) was recorded. Head emergence was defined as the moment at which the head extends far enough from the body so that the eyes protrude past the anterior margin of the carapace. If an individual extended its head immediately after being placed on the ground, it was given a head emergence latency time of zero seconds, and trials in which individuals did not emerge from their plastron after 600 seconds were terminated. Individuals were categorized as bold if head emergence latency time was less than or equal to ten seconds, while individuals were listed as shy with times greater than ten seconds. The cutoff time

for bold vs shy classification was determined from previous strongly bimodal results in Eastern Box Turtle head emergence assays [see 32,34]. Due to logistical constraints from concurrent research activities, this test was only performed once on each individual. However, several previous papers establish that the behavioral assay used is highly consistent in these turtles, making this behavioral measurement a reasonable proxy for personality [32–34,44]. In addition, previous studies (i.e., [44]) have also used a single measurement and justified this given the previously-reported repeatability of the behavioral assay.

During microbiota sampling, the cloaca, skin, and oral cavity of each encountered individual was swabbed for 30 seconds with a sterile Puritan Rayon-tipped swab (Puritan Medical Products, Guilford, ME, USA) and stored in 95% ethanol in a 1.7mL collection vial. All tools were sterilized prior to use and sterile gloves were worn during handling. After collection, samples were stored at −80°C until DNA extraction. After turtle capture and completion of behavioral assays (see below), sex was determined based on secondary characteristics (i.e., the shape of the plastron, the position of the cloaca in relation to the tail, and coloration) and recorded as covariate data. Immature individuals were marked as "unknown" sex.

### DNA extraction and sequencing

Microbiota DNA was isolated from oral, cloacal, and skin swabs using a commercially available kit (QIAmp DNA Blood Mini Kit and DNAeasy kit, Qiagen, Valencia, CA, USA). Protocols from the manufacturer kit were closely followed. DNA was assessed for purity and concentration using UV-Vis spectrophotometry (NanoDrop 1000, Thermo Fisher Scientific, Waltham, MA, USA) and DNA was stored at -80°C. Bacterial amplicons were then amplified via Polymerase-Chain-Reactions (PCR) using universal 16s V4 rRNA primers. For the bacterial 16S rRNA gene, primers 515F and 926R were used to create amplicons [45]. PCR products were validated using agarose gel electrophoresis. After high-quality DNA had been extracted and amplified, amplicons were sequenced on a P1 600cyc NextSeq2000 (Illumina) Flowcell to generate 2x301bp paired end (PE) reads.

### Bioinformatics and statistical analyses

The relationship between Eastern Box Turtle microbiota and boldness levels were investigated with QIIME2 [46]. The QIIME2 Atacama soil microbiome tutorial was modified to import, trim and denoise sequence data. Forward and reverse reads were combined via Casava-paired end. Sample types were separated, and sequence data were analyzed following the QIIME2 Parkinson's Mouse tutorial [46]. Faith's phylogenetic diversity, Pielou's evenness, Shannon's Diversity and Observed Features were used to measure alpha diversity. Single-factor Kruskal-Wallis tests were used to compare alpha diversities. ANOVAs were used to investigate interacting effects. Weighted and unweighted UniFrac distances, Bray-Curtis distance, and Jaccard Distance were used to measure beta diversity. The factors that had potential influence on the microbiota of bold/shy turtles were analyzed with a permutational multivariate analysis of variance (PERMANOVA) [47]. Taxonomic assignment into OTUs was achieved using a Greengenes 99% classifier (ver 2022.10 [48]). Finally, taxonomy bar charts were created to compare within and between sample groups.

Our taxonomy and feature table files were combined into.biom files. In addition, our cloacal, oral, and skin metadata tables were edited to be compatible with MicrobiomeAnalyst [49]. Using the.biom and metadata files, each sample type (oral, cloacal, and skin) was individually investigated for alpha diversity (tested using Chao1 and Shannon's Diversity), beta diversity (tested using NDMS and PERMANOVA), composition of the core microbiome, and Linear discriminant analysis Effect Size [50]. Specifically, LEFse was used to evaluate OTUs (Operational Taxonomic Units) that differed significantly in abundance in the context of boldness, and the effect size of differential abundance features.

We used PICRUSt for functional prediction [51]. Within these analyses, a second LEFse was performed along with KEGG-style association analyses to identify significant OTUs, metabolic pathways, and modules in the context of personality. For all analyses, default parameters were used.

## Results

Nine cloacal samples, ten oral samples, and nine total skin samples were collected from 11 turtles. These included 9 adults (six males and three females), one juvenile, and one subadult. At least one sample type (oral, cloacal, and skin) was collected from each turtle, however one or more sample types were not collected from three turtles (1044, 2002, and 30) due to post sampling failure. Six turtles were classified as bold and five were classified as shy. Skin, oral, and cloacal swabs were analyzed separately to avoid violating assumptions of independence (and to gain an understanding surrounding microbiota compositions of these specific areas on and within Eastern Box Turtles). 2,179,842 raw cloacal reads, 1,956,538 raw oral reads, and 2,024,209 raw skin reads were produced and successfully demultiplexed. After filtering,1,181,837 cloacal reads (mean per individual: 131,315), 361,292 oral reads (mean per individual: 36,129), and 804,272skin reads (mean per individual: 89,364) remained.

### Alpha diversity

In this study, microbiota alpha diversity is the species diversity *within* a sample or sample type, e.g., "cloacal" *or* "oral" *or* "skin". The alpha diversity is calculated for each individual swab and the averages of these are then compared for alpha correlations. Chao1 and Shannon's Diversity Index were used to quantify the alpha diversities and investigate the relationship between personality classification (bold/shy) and microbiota diversity differences within sample types (oral, cloacal, and skin). No significant difference in alpha diversities were found between personality types (p > 0.05, Fig 1).

In each individual sample type (cloacal, oral, and skin) there was no significant difference in alpha diversity of the microbiota between sexes and/or personality types (p > 0.05). Next, in order to compare whole-animal microbiomes, we pooled sequence data from the three sample types (cloacal, oral, and skin) as available. In this case, there was a significant difference in the microbiota composition between turtles of different sexes, specifically males (n = 6) and unknown sex (n = 2; the subadult and juvenile turtles) (Kruskal-Wallis pairwise test, p = 0.045500, S1 Fig). There was also a significant interaction between personality type and sex found using ANOVAs on Faith's phylogenetic diversity (PR(>F) =0.0109), Pielou's evenness (PR(>F) = 0.000989), Shannon's diversity index (PR(>F) = 0.00397) and Observed Features (PR(>F) = 0.0372) (S2 Fig). Pairwise t-tests were performed for ANOVAs on Faith's phylogenetic diversity, Pielou's evenness, Shannon's Diversity and Observed Features (S1 Table).

### Beta diversity

Microbiota beta diversity is the microbial community differences *among* sample types, e.g., "bold" *and* "shy". Interactions between beta diversity between personality and sample type (cloacal, oral, and skin) were investigated using NMDS and PERMANOVA (Bray-Curtis Index method).

PERMANOVAs were used to test for significant beta diversity using weighted and unweighted UniFrac distance, Jaccard Distance, and Bray-Curtis distance. No significant differences were found in any diversity tests between bold/shy personalities within any sample type (skin, oral, cloacal, and combined) (p > 0.05, Fig 2; see S3–S5 Fig for Jaccard and UniFrac distance).

### Taxonomy

Relative abundances of microbial phyla are shown in Fig 3. Within cloacal samples, Actinobacteria dominated both bold groups (71.48%) and shy groups (68.39%). In bold cloacal samples, firmicutes comprised the second-most abundant phyla (13.07%), followed by Proteobacteria (8.48%), while in shy cloacal samples Thermi (9.80%), followed closely by Firmicutes (8.27%), and Proteobacteria (8.06%) comprised the second, third, and fourth-most abundant phyla. Actinobacteria also dominated bold (75.16%) and shy (84.22%) groups in oral samples, followed by Bacteroidetes in both groups (bold: 14.84%; shy: 6.17%). In shy oral samples, Firmicutes (5.23%) closely followed Bacteroidetes. Actinobacteria once

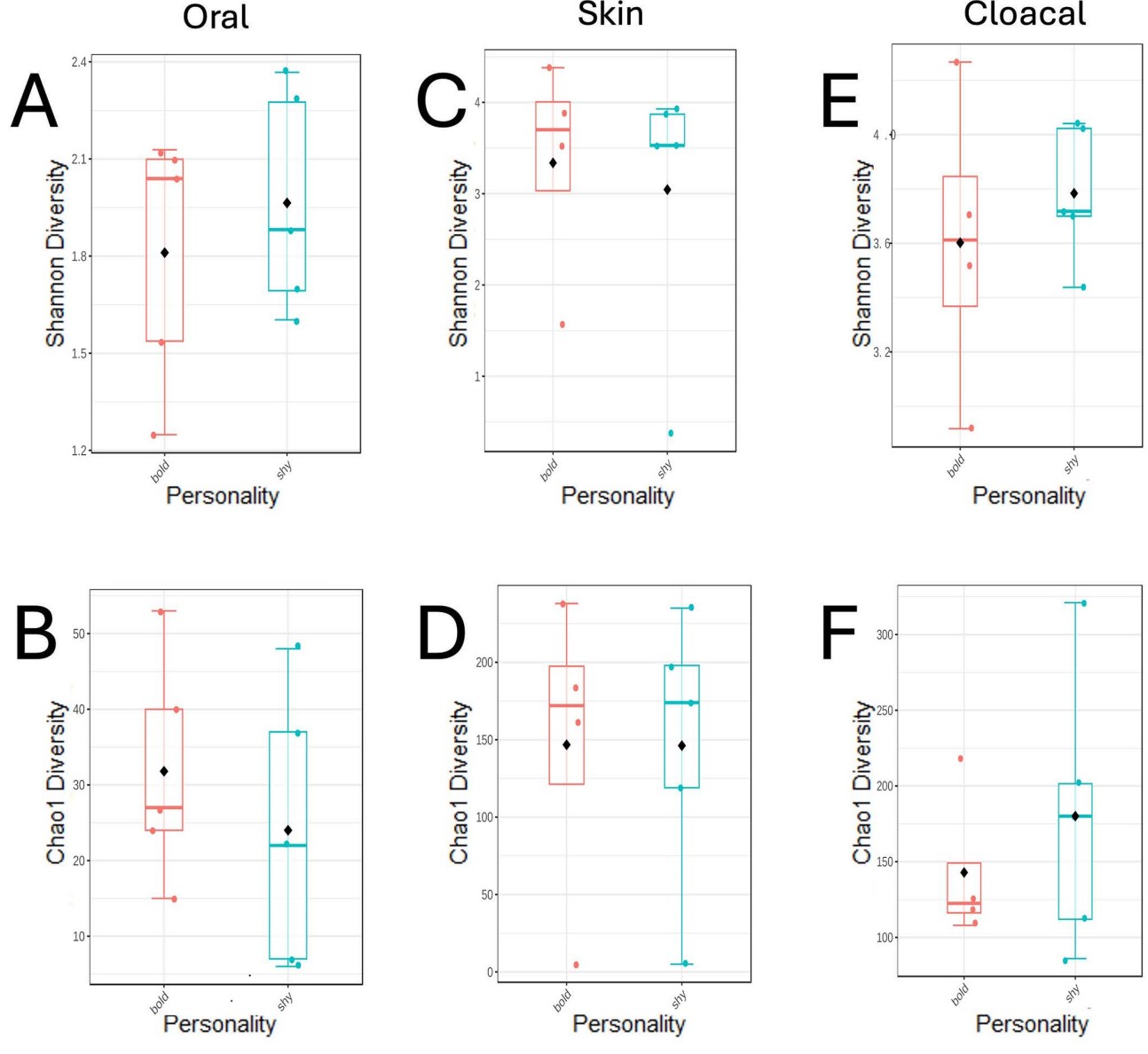

**Fig 1. Oral, skin, and cloacal alpha diversities of bacterial communities in bold and shy hosts.** Shannon diversities are given in (A), (C), and (E;) Chao1 diversities are given in (B), (D), and (F). Oral diversities are given in (A) and (B); skin diversities are given in (C) and (D); cloacal diversities are given in (E) and (F). Kruskal-Wallis tests for differences between host personality were not significant for any comparison (p > 0.05).

again dominated both bold (79.02%) and shy (80.03%) groups, followed by Thermi in both groups (bold: 13.85%; shy: 15.19%, Fig 3B) (see S2 and S3 Tables for all phyla abundance values).

## Core microbiome

We found 58.8% composition was shared between personalities, with shy turtles possessing a richer microbiome than bold turtles (Fig 4).

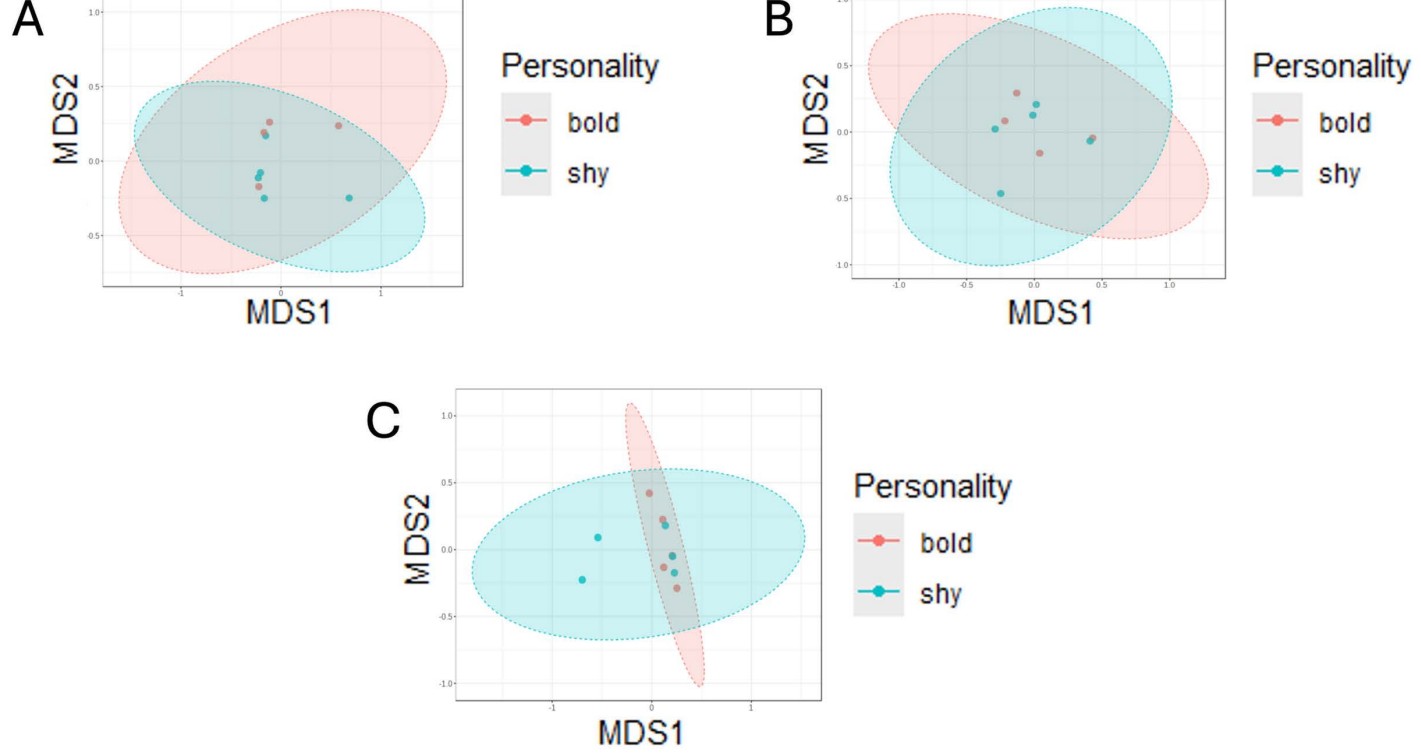

**Fig 2. Non-metric multidimensional scaling of Bray-Curtis beta diversity of bacterial communities among bold (N = 6) and shy (N = 5) individuals. (A)** skin samples ($R^2$ = 0.091935, p = 0.567), **(B)** Oral samples ($R^2$ = 0.11295, p = 0.463), and **(C)** cloacal samples ($R^2$ = 0.095808, p = 0.851). Shaded ellipses represent the 95% confidence interval.

Shared taxa within the core microbiome of each sample type were assessed at 0.01–0.076% relative sample abundances. Skin and oral samples showed no taxa shared by at least 50% of turtles. However, cloacal samples revealed five taxa shared at 60% sample prevalence (Table 1) and one significant taxon at 80% prevalence.

### LEFse analyses

Four significantly differentially abundant OTUs were found between bold and shy turtles (Fig 5), including one Alphaproteobacteria, two Actinobacteria, and one [Thermi].

### Functional predictions

For all turtles, four significantly overrepresented metabolic pathways were identified in oral samples, two in cloacal samples, and none in skin samples (Table 2). Six significantly overrepresented modules were found in oral samples, three in cloacal samples, and two in skin samples (Table 3). The purine degradation, xanthine→urea module was identified in both oral and cloacal pathways. All other statistically overrepresented pathways and modules were unique to each sample type. None were significantly different between bold and shy.

### Discussion

This study aimed to: (1) characterize the endangered, long-lived, and non-model Eastern Box Turtle microbiome, and (2) increase understanding of the relationship between Eastern Box Turtle microbiome composition and indicators of bold vs

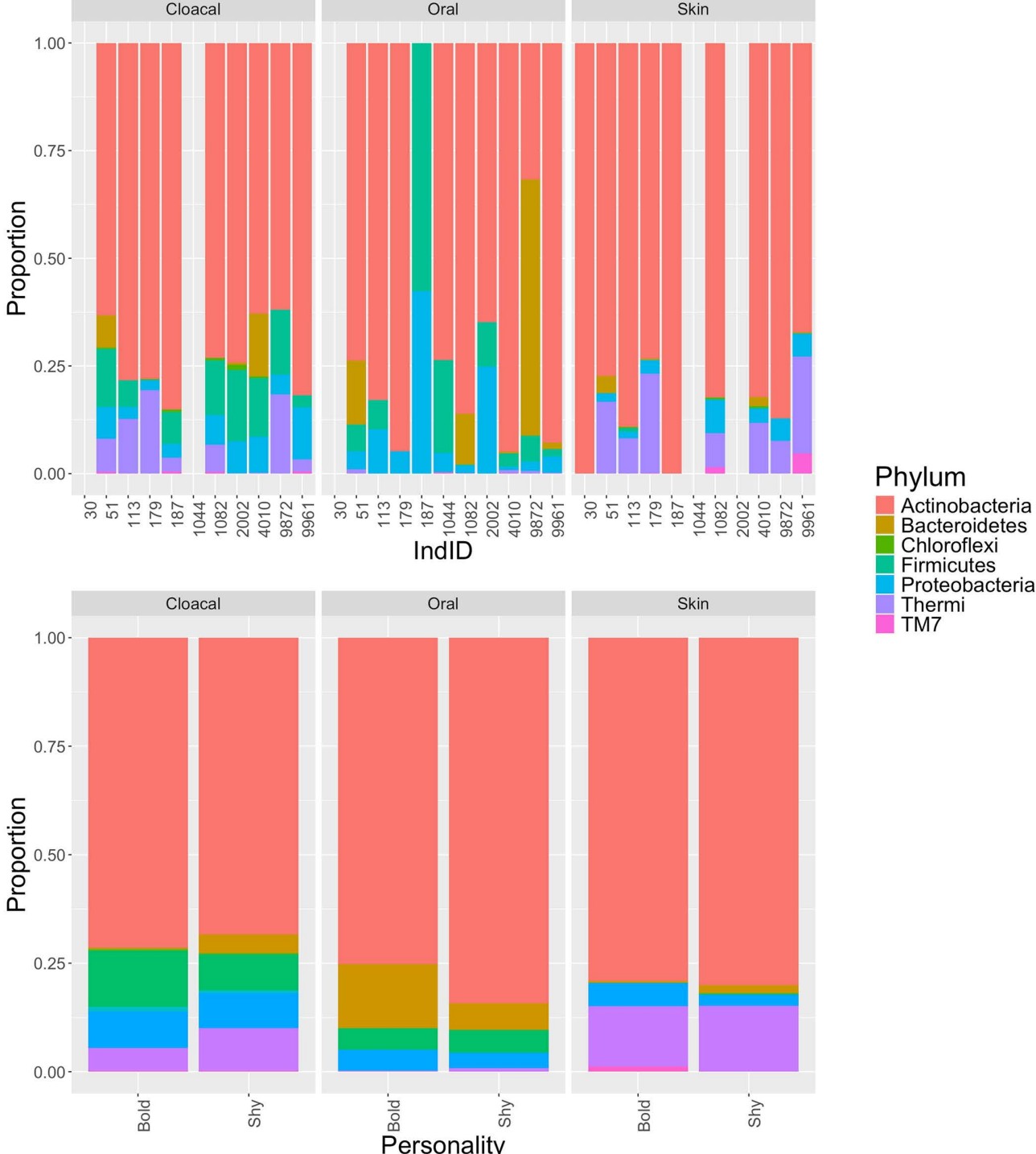

**Fig 3. Relative abundance at the phylum level for cloacal, oral, and skin samples from bold and shy turtles.** (A) Relative abundance per individual, separated by sample type and bold vs shy classification. (B) Averaged relative abundance for bold vs shy individuals, separated by sample type.

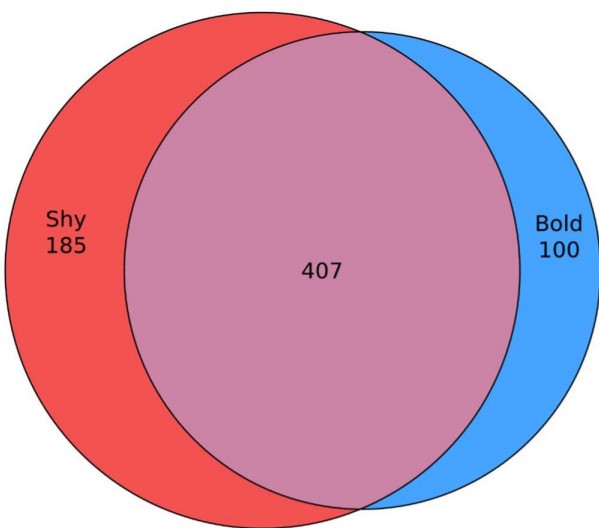

**Fig 4. Unique and shared taxa (to species level) among bold and shy turtles.**

**Table 1. Core cloacal microbiome taxa.**

| Taxonomic rank | | | | | | | |
|---|---|---|---|---|---|---|---|
| **Kingdom** | **Phylum** | **Class** | **Order** | **Family** | **Genus** | **Species** | **Prevalence across all turtles** |
| Bacteria | Actinobacteria | Actinobacteria | Actinomycetales | Gordoniaceae | *Millisa* | *brevis* | 80% |
| Bacteria | Proteobacteria | Betaproteobacteria | Burkholderialies | Alcaliginaceae | NA | NA | 60% |
| Bacteria | Bacteroidetes | Flavobacteria | Flavobacteriales | [Weeksellaceae] | NA | NA | 60% |
| Bacteria | Proteobacteria | Betaproteobacteria | Neisseriales | Neisseriaceae | NA | NA | 60% |
| Bacteria | Actinobacteria | Actinobacteria | Actinomycetales | Intrasporangiaceae | NA | NA | 60% |

shy personality types. We predicted that the microbiome composition of bold and shy turtles would be significantly different. Specifically, we predicted that alpha diversity would be significantly higher in bold turtles, and that beta diversities would significantly differ between them. We also predicted that bold turtles would have metabolic pathway activity that promotes increased metabolism and aggression, and we expected shy turtles would have significantly higher metabolic pathways that promote chemical messengers associated with depression.

The separated microbiota of skin, oral, and cloacal samples collected from bold and shy Eastern Box Turtles from Indiana showed some differences (Figs 1–4) but did not differ significantly in alpha or beta diversity. Shy turtles in this sample possessed a richer microbiota than bold turtles (Fig 4). Similarly, [23] found no significant differences in alpha diversity between bold and shy individuals. However, [24] found bold Tibetan macaques to have significantly lower alpha diversity indices than shy macaques. The authors suggest that as bolder macaques tend to be more aggressive, the resulting heightened levels of cortisol can lead to increased aggression and higher levels of stress, indirectly causing a decrease in gut microbiota diversity. Likewise, previous studies have shown bolder Eastern Box Turtles to have (non-significant) higher corticosterone (the reptilian analogue of primate cortisol [52]) than shy turtles, perhaps suggesting that although our diversity results were not significant, the increased richness observed in shy turtles could be a result of lower stress levels due to engaging in less risky behavior. An increased sample size is needed to thoroughly investigate this hypothesis.

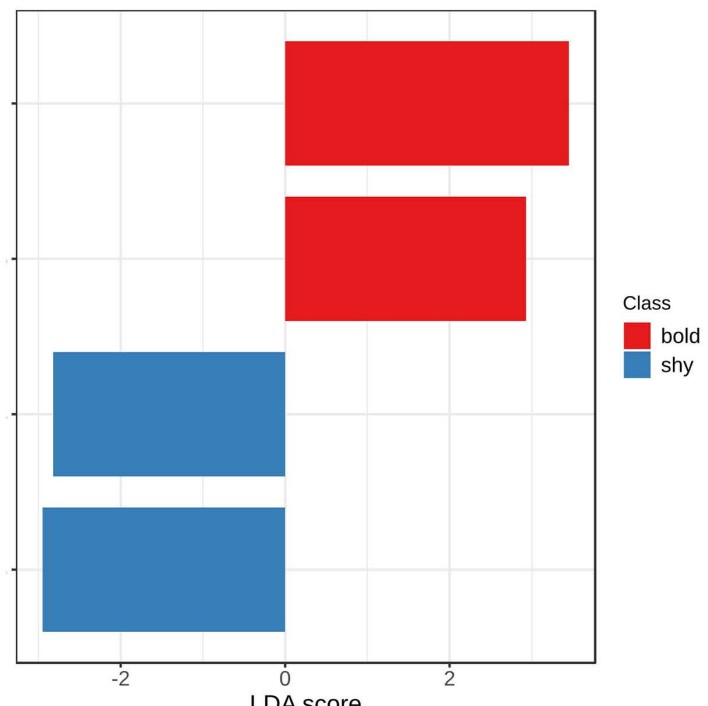

k__Bacteria; p__Proteobacteria;
c__Alphaproteobacteria; o__Rhodobacterales;
f__Rhodobacteraceae; g__Albidovulum;
s__inexpectatum

k__Bacteria; p__Actinobacteria; c__Actinobacteria;
o__Actinomycetales; f__Dermatophilaceae

k__Bacteria; p__Actinobacteria; c__Actinobacteria;
o__Actinomycetales; f__Nocardioidaceae;
g__Nocardioides

k__Bacteria; p__[Thermi]; c__Deinococci;
o__Deinococcales; f__Deinococcaceae;
g__Deinococcus

**Fig 5. Significantly (p < 0.05) differentially abundant OTUs.**

**Table 2. Significantly overrepresented metabolic pathways identified in each sample type.**

| Sample Type | | |
|---|---|---|
| *Oral* | **Pathway** | **Significance** |
| | Steroid biosynthesis | p = 0.0329 |
| | Primary bile acid biosynthesis | p = 0.0431 |
| | Naphthalene degradation | p = 0.0441 |
| | Retinol metabolism | p = 0.0467 |
| *Cloacal* | **Pathway** | **Significance** |
| | Atrazine degradation | p = 0.0068 |
| | Chlorocyclohexane and chlorobenzene degradation | p = 0.0328 |

No significant pathways were identified for skin samples. Alpha value = 0.05.

Our alpha diversity analyses indicate a significant difference only in the combined (skin, oral, cloacal) microbiota composition between turtles of different sexes. From these tests, the significant difference seems to be between males (higher alpha diversity) and turtles of unknown sex (lower alpha diversity), while females show no significant difference between these two groups (S1 Fig). Our alpha diversity ANOVA analyses suggest a significant effect on both sex and personality on combined (skin, oral, and cloacal) microbial alpha diversity in turtles (p < 0.05; S1 Table). Our data suggest female turtles have a significantly higher microbial diversity than male turtles, and shy turtles have a significantly higher microbial diversity than bold turtles (S2 Fig). Similarly, when comparing the gut microbiome of adult male and female Chinese alligators (*Alligator sinensis*), [53] found no significant difference in alpha diversity of the gut microbiota, but did

**Table 3. Significantly overrepresented metabolic modules identified in each sample type.**

| Sample Type | | |
|---|---|---|
| *Oral* | **Module** | **Significance** |
| | Purine degradation, xanthine→urea | p=0.0404 |
| | Cysteine biosynthesis, homocysteine+serine→cysteine | p=0.0158 |
| | Lysine biosynthesis, AAA pathway, 2-oxoglutarate→2-aminoadipate to lysine | p=0.0325 |
| | Trans-cinnamate degradation, trans-cinnamate→acetyl-CoA | p=0.0366 |
| | Semi-phosphorylative Entner-Doudoroff pathway, gluconate/galactonate→glycerate-3P | p=0.0384 |
| | Homoprotocatechuate degradation, homoprotocatechaute→2-oxohept-3-enedioate | p=0.0498 |
| *Cloacal* | **Module** | **Significance** |
| | Purine degradation, xanthine→urea | p=0.0036 |
| | Cytochrome c oxidase | p=0.0086 |
| | Malonate semialdehyde pathway, propanoyl-CoA→acetyl-CoA | p=0.0123 |
| *Skin* | **Module** | **Significance** |
| | Glutathione biosynthesis, glutamate→glutathione | p=0.0256 |
| | Catechol ortho-cleavage, catechol→3-oxoadipate | p=0.0356 |

Alpha value=0.05.

find a significant interaction between alpha diversity of juvenile Chinese alligators. In that study, juvenile females had a higher alpha diversity than juvenile males. Additionally, juvenile males had lower sex-biased microbial phyla and genera but an increase in *Lactobacillus.* This may have been due to the different environments (diet, pathogenic bacteria, etc.) and hormonal development of male vs. female juveniles [53]. In striped plateau lizards (*Sceloporus virgatus*), the alpha diversity changed differently throughout the breeding season between males and females. Similar to our study, females demonstrated a generally higher richness, which may be due to behavioral differences such as energy allocation and feeding habits [54]. Our results may be due to a number of factors, including time of year. Past studies [e.g., 54] found the lowest microbial diversity during the breeding season and our samples were collected during the beginning of the breeding season, which may be why there was no significant difference between sexes. However, it is important to note we had an uneven sex and age ratio with the turtles we were able to collect for this study. Additional research with larger and even sex ratio size is needed to better understand the microbiome relationships among personality, age, time of year, sex of turtles, and if there is an interaction between variables.

## Microbiome composition

Actinobacteria, Proteobacteria, and Firmicutes dominated microbiota composition in all three sample types but in different proportions (Table 1). Proteobacteria and Firmicutes, as well as Bacteroidetes, have been reported in other turtle studies as being the most detected phylum, which includes Krefft's river turtle (*Emydura macquarii krefftii*), green sea turtles (*Chelonia mydas*), and Loggerhead sea turtles (*Caretta caretta*) [55–57]. In these studies, Actinobacteria was also present but in lower abundance [38,55,56]. These results are not surprising, as previous studies have found Firmicutes to be one of the major phyla that comprise various vertebrate gut microbiomes [58–60] and Proteobacteria are known to have significant effects on gut microbial processes [59]. In addition, Actinobacteria are commonly found in soil and aquatic habitats and function to degrade various organic substances, such as polysaccharides and protein fats [61]. Actinobacteriahave been found to form symbioses with vertebrates to facilitate intake of nutrition, detoxification of specific compounds, growth performance, and protection from pathogens [61]. A previous study observed a significant increase in Firmicutes and Actinobacteria as well as an increase in exploratory behavior by gut-microbiome-altered mice [62]. As exploratory behavior

has previously been correlated with bold personality types [63], this increase in exploration may be indicative of increased boldness, and this boldness may be associated with increases in Firmicutes and Actinobacteria.

Bacteroidetes were also relatively abundant, specifically in oral samples. Bacteroidetes are known to dominate gut flora [64], regulate functions of the immune system, and break down carbohydrates, thus facilitating digestion and nutrient acquisition [65, 66]. One study, after transferring fecal microbiota from human patients with major depressive order to mice, suggested that Bacteroidetes cause a disturbance in serotonin metabolism, resulting in decreased serotonin levels and enhanced depression susceptibility. This, in turn, is thought to create stress-induced, abnormal behavioral patterns in mice [67]. Bacteroidetes were significantly higher in bold gerbils [23] and though were not significantly different here, were more abundant in oral and skin swabs of bold turtles (Fig 3).

Proteobacteria play a role in the biosynthesis of vitamins and contribute to complex sugar breakdown and fermentation [23]. Two Proteobacteria OTUs were found in the core microbiome of cloacal samples (Table 1), and one significant Proteobacteria OTU was found enriched for bold turtles in cloacal LEFse analyses (Fig 5). This suggests a potentially critical role in bold personality type, although larger sample sizes are needed to definitively conclude the importance of Proteobacteria in personality of Eastern Box Turtles.

When analyzing cloacal data, four significantly overrepresented OTUs were found, including two showing higher prevalence in bold turtles and two in shy (Fig 5). *Albidovulum inexpectum*, overrepresented in bold turtles, was first isolated from marine hot springs [68] and while it has not previously been associated with animal behavior, other taxa in its class (Alphaproteobacteria) have been associated with depression in humans [69], suggesting a possible gut-brain axis connection. The family Dermatophilaceae was also overrepresented in bold turtles. Though no genus or species were identified, members of this family have been associated with human and other animal skin flora and are sometimes pathogenic [70]. The genus *Nocardioides* was overrepresented in shy turtles, and has been described elsewhere in carp [71] and in Yangtze river water [72], where they were significant predictors of source location. The genus *Deinococcus* was also overrepresented in shy turtles. Members of this phylum are extremophiles, resistant to multiple stressors (e.g., radiation, desiccation, and oxidizing agents) [73] and their association with animal personality has not previously been made.

We found evidence for a small core microbiome of shared cloacal taxa from bold and shy turtles (Table 1). These taxa, including Proteobacteria and Bacteroidetes, comprise common phyla for turtle gut microbiota. However, Actinobacteria has not been core in these studies [55–57]. In green sea turtle cloacal studies, Firmicutes and Bacteroidetes were found to be the dominant phyla [55,57,74], which was not the case here. With an increased sample size there might be more similarities shared within the core microbiome.

## Prominent metabolic pathways and modules

As boldness and aggressiveness have been previously correlated [75], we predicted that bold-type individuals would have significantly higher rates of steroid biosynthesis. Steroids can increase sex drive, and an increase in steroid biosynthesis has been found to be associated with increased aggression in mice [76]. The steroid biosynthesis metabolic pathway found to be overrepresented in turtles in the current study may suggest effects of diet [77], stress [78], or sex expression [79], but given this wide diversity of possible effects warrants further investigation.

The purine degradation→urea module was the only significant module found in two sample types (oral and cloacal, Table 3), potentially indicating its critical role in the assignment of personality type. In an experiment that exposed tadpoles to increased amounts of urea, activity levels decreased [80], which could be associated with shy-type personality. Thus, increases in urea may contribute to characterization of the shy personality type. We found little to no research on the effects of urea and behavior on reptiles; additional research would be beneficial in understanding if there is an influence.

A primary bile acid biosynthesis pathway was found to be significant in oral samples (Table 2). This finding agrees with previous research in lab-reared mice, which found that mice deficient in a prominent enzyme in the bile acid synthesis

pathway displayed behavioral irregularities, one major reason being the lack of vitamin D and E3 produced because of the enzyme deficiency [81]. Past studies have linked vitamin D deficiency to problems in activating serotonin, which in turn can contribute to the development of depression [82]; higher levels of depression have been suggested to occur in individuals characterized by shy personality [24]. Although the presence of depression has not been fully characterized in turtles (but see [83]), one can hypothesize that reductions in bile acid biosynthesis may lead to reduced serotonin levels, which may play a role in the characterization of shy personality types.

## Conclusion

Using oral, skin, and cloacal swabs, we investigated the relationships between bold and shy behavior types in Eastern Box Turtles encountered at three sites in West-Central Indiana. Only when sample types (oral, cloacal, and skin) were combined, were there any significant differences, specifically in males and turtles of unknown sex. We also found a significant difference between the microbiota of male and female Eastern Box Turtles, as well as bold and shy turtles in our alpha diversity ANOVA analyses. However, beta diversity between bold and shy Eastern Box Turtle individuals across three sample types (oral, cloacal, and skin) were not significantly different. We found five OTUs that were present in at least 50% of individual cloacal samples (Table 1) and four significantly overrepresented cloacal OTUs in both bold and shy turtles (Fig 5). We found several statistically significant pathways and modules across sample types (Tables 2 and 3) that may be related to personality. Additionally, our results suggest potential prominent metabolic pathways, modules, and OTU's that are implicated in behavior of Eastern Box Turtles and thus partial determinants of bold vs shy personality types. However, our conclusions should be considered in the context of the limitations of our sample size and ratio of male, female, and subadult and juvenile captured turtles. This study contributes to a small but growing understanding of the Eastern Box Turtle microbiome, and is among the first to relate turtle microbes to behavior. Future studies should increase sample size and geographic range to further explore the associations between the gut microbiome and behavior. In addition, future research should continue to broaden the depth of knowledge of wild and reptilian microbiome-behavior associations to increase understanding of the effect of microbiota on individual and ecological processes and mechanisms, and thus the role that microbiome composition can play in management of these highly threatened populations.

## Supporting information

**S1 Fig. Alpha diversity (Faith's phylogenetic diversity) of bacterial communities between sexes.** Males (n = 6) and n.a. (n = 2) (Kruskal-Wallis pairwise test, p = 0.045500).
(DOCX)

**S2 Fig. ANOVA (Faith's phylogenetic diversity) of bacterial communities between bold and shy turtles to sex. sum_sq = 2186.8, df = 1, F = 113.191, PR(>F) = 0.0109.**
(DOCX)

**S3 Fig. Non-metric multidimensional scaling of Jaccard beta diversity of bacterial communities among bold (N = 6) and shy (N = 5) individuals.** (A) skin samples (R2 = 0.11447, p = 0.595), (B) Oral samples (R2 = 0.094497, p = 0.734), and (C) cloacal samples (R2 = 0.1137, p = 0.894). Shaded ellipses represent the 95% confidence interval.
(DOCX)

**S4 Fig. Principal Coordinates Analysis of Weighted Unifrac beta diversity of bacterial communities among bold (N = 6) and shy (N = 5) individuals.** This includes skin samples (p = 0.595), oral samples (p = 0.734), and cloacal samples (p = 0.8).
(DOCX)

**S5 Fig. Principal Coordinates Analysis of Unweighted Unifrac beta diversity of bacterial communities among bold (N = 6) and shy (N = 5) individuals.** This includes skin samples (p = 0.595), Oral samples (p = 0.734), and cloacal samples (p = 0.894).
(DOCX)

**S1 Table. ANOVA results on Bold/Shy and Sex (male, female, and na).** Faith pd: Bold_or_Shy:Sex, sum_sq (4632.50749), df (2.0), F (16.572713), PR(>F) (0.004743); Evenness: Bold_or_Shy:Sex, sum_sq (0.019633), df (2.0), F (37.233848), PR(>F) (0.00049); Shannon: Bold_or_Shy:Sex, sum_sq (14.199353), df (2.0), F (23.234192), PR(>F) (0.001921); Observed features: Bold_or_Shy:Sex, sum_sq (688441.626667), df (2.0), F (8.660089), PR(>F)(0.021627). Pairwise t-tests were performed for Faith pd (A), Evenness (B), Shannon (C), and Observed features.
(DOCX)

**S2 Table. Relative abundances of microbial phyla by sample-type and personality.**
(DOCX)

**S3 Table. Relative abundances of microbial phyla by individual turtle.**
(DOCX)

## Acknowledgments

We are grateful for the assistance in fieldwork collection by Thomas Kay, Daren Glore, Jacob Penrose, and Bradley Johnson.

## Author contributions

**Conceptualization:** Bradley E. Carlson.

**Data curation:** Steven J.A. Kimble.

**Formal analysis:** Kaija Harlow, Elizabeth K. Service, Jace E. Geiger, Steven J.A. Kimble.

**Funding acquisition:** Bradley E. Carlson, Steven J.A. Kimble.

**Investigation:** Bradley E. Carlson.

**Methodology:** Bradley E. Carlson.

**Project administration:** Bradley E. Carlson.

**Resources:** Bradley E. Carlson, Steven J.A. Kimble.

**Supervision:** Bradley E. Carlson, Steven J.A. Kimble.

**Validation:** Jace E. Geiger, Bradley E. Carlson.

**Visualization:** Kaija Harlow, Elizabeth K. Service, Steven J.A. Kimble.

**Writing – original draft:** Kaija Harlow, Elizabeth K. Service.

**Writing – review & editing:** Jace E. Geiger, Bradley E. Carlson, Steven J.A. Kimble.

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
