## [Decision Letter · Decision Letter 0]

6 Oct 2025

PONE-D-25-40604The Relationships Between Box Turtle Gut Microbiomes and PersonalityPLOS ONE

Dear Dr. Harlow,

Thank you for submitting your manuscript to PLOS ONE. After careful consideration, we feel that it has merit but does not fully meet PLOS ONE’s publication criteria as it currently stands. Therefore, we invite you to submit a revised version of the manuscript that addresses the points raised during the review process.

We look forward to receiving your revised manuscript.

Kind regards,

Bernd Schierwater, Ph.D

Academic Editor

PLOS ONE

Journal Requirements:

2. To comply with PLOS ONE submissions requirements, in your Methods section, please provide additional information regarding the experiments involving animals and ensure you have included details on (1) methods of sacrifice, and (2) efforts to alleviate suffering.

3. Please note that funding information should not appear in the Acknowledgments section or other areas of your manuscript. We will only publish funding information present in the Funding Statement section of the online submission form. Please remove any funding-related text from the manuscript. 

Reviewers' comments:

Reviewer's Responses to Questions

**Comments to the Author**

1. Is the manuscript technically sound, and do the data support the conclusions?

Reviewer #1: Yes

Reviewer #2: Partly

2. Has the statistical analysis been performed appropriately and rigorously? 

Reviewer #1: Yes

Reviewer #2: No

3. Have the authors made all data underlying the findings in their manuscript fully available?

Reviewer #1: Yes

Reviewer #2: Yes

4. Is the manuscript presented in an intelligible fashion and written in standard English?

Reviewer #1: Yes

Reviewer #2: No

5. Review Comments to the Author

Reviewer #1: The authors have created a very interesting, detailed, and well-written manuscript with the commendable objective of providing novel information to assist in the management and protection of the threatened Eastern Box Turtle species.

Although the sample size is quite limited, this is understandable given the species studied. It is also appreciated that the discussion acknowledges the need for increased sample size overall and the need to balance sexes in future studies.

The paper is acceptable, however there are several minor notes:

The references section does not include numbers that would correspond to the in-text citations. In addition, page numbers are missing.

It is stated that a total of 12 turtles were collected but only 11 total turtles deemed bold/shy (first mentioned in Figure 1) with no mention of the 12th. It could be confused that this was either the turtle that supposedly did not emerge after 600 seconds where the trail was terminated (is it not mentioned if this turtle was included with the list of shy individuals or not), or the immature unknown sex individual for sex comparison studies, although only the 10 adults had their sexes determined; so this could be more clearly stated. In addition, cloacal, oral, and skin samples were not taken from all 12 turtles (only either 10 or 11) with no apparent mention for this reason, even though it is mentioned that samples were taken from each encountered individual.

In regards to Figure 2: Is there a reason only the Bray-Curtis distance analysis is shown in a figure and not the other tests for beta diversity? Additionally, this figure may work better if “Skin”, “Oral”, and “Cloacal” were shown by their respective A-C designated letters as was included in Figure 1. The text also includes “combined” as a sample type for this test which does not appear in the figure. Lastly, the ellipses are not “shaded”.

Side Note: Although not required, consider en dashes for ranges, and not starting a sentence directly with a bracketed citation.

Reviewer #2: This interesting study investigates the relationship of Eastern Box Turtle behaviour and their microbiome, on a sample of 12 turtles, collected from several testing sites and tested in a time-to-head-emergence essay that was linked with microbiome analyses from the turtles’ cloacal, skin and oral swabs. The Authors found that females and shy turtles had significantly higher alpha microbiome diversity than males and bold turtles, and there was no significant difference in beta diversity. The Authors also found several overabundant microbial taxa, pathways and modules that differed between bold and shy turtles. The steroid biosynthesis pathway was overrepresented in bold turtles, whereas bile acid biosynthesis pathway and purine degradation to urea module were overrepresented in shy turtles. While the results are overall interesting, I could not identify the main “take home messages” and conclusions of this manuscript. There are many aspects of the manuscript that should be improved before it can be considered for publication, and I list these major and minor point below, hoping they will be helpful for the Authors in revising their manuscript.

Major points.

Labelling throughout the manuscript should be improved. As the Authors did not conduct repeated testing (L181), labelling behaviours as ‘personality’, ‘boldness’ or ‘shyness’ is misleading, instead of using ‘personality’ as a term, the Authors should use ‘behaviour’. Furthermore, why do Authors make a distinction between “microbiome” and “microbiota” (L76-78) and then use them interchangeably in the text?

Was the overall aim of the manuscript just to get more data on Eastern Box Turtles because they are “reptiles and long-lived, non-model organisms” (L28) or that “and that they were used because of “the dire need for more data on their behaviour and microbiome as reptiles and non-model organism” (L15-151). This is a bit weak reason.

Methods and data collection are unclear. For instance, it seems that some turtles were already captured and tracked beforehand (“many turtles encountered in this study had been previously captured and tracked” L157-158; “some turtles” L164), but how many turtles were captured of the ones tested in this manuscript is not precisely given. Further, how were the tests done, was the “time-to-head-emergence” test conducted first, then the measurements, then the swabs or did you follow other order of data collection? How were the turtles located? What motivated your decision to label turtles “bold” if they emerged in less than 10 and “shy” in more than 10 seconds – was there any biological reason for this? Why did the Authors conduct only one behavioural test, when they knew that repeated testing is needed to assess personality?

Statistical analyses should be carefully conducted and results discussed with taking a small sample size that the Authors collected in mind (only 12 turtles, uneven number of females and males, comparisons with differing numbers e.g. 7 versus 2 animals, etc).

Statistical details should not be reported under Figure and Table descriptions, but instead in the main text of the manuscript.

The analyses conducted with linking behaviour and microbiome are not very clearly described, so the reader will be left somewhat puzzled what was done where. Do these analyses refer to L210-212? Which analyses were done to check the effect of personality, sex, age etc on microbiome? L250. What kind of interaction was found in terms of personality type & sex? L333-334. Here it seems that sex effects are convoluted with age, as IDs of unknown sex were also the juvenile and subadult ones? Please clarify.

Literature coverage was made somewhat superficially, without giving specific examples. This is particularly true for Abstract (very generally written without discussion and conclusion sentences, giving general claims on phenomena that are quite underexplored; e.g. L21-22, L23-24, L24-25, L44-45). In general, details of previous studies are missing from the manuscript.

Related point is that it is not clear from the Discussion what the Results represent and why they are important.

Conclusion and overall messages that the reader can get from the manuscript are not clearly conveyed.

References are very messy and unnumbered which makes it difficult to follow numbering in text. There is no consistent style in capitalization of words within a title. Editor names are given for journal articles which is not needed, links that are not doi identifiers, species names are not in italic, journal names are not capitalized etc.

Minor points.

L50-51 is redundant with L61-64.

L54. “to explain variation in personality” can be omitted.

L135. What does it mean that “the microbiomes will have significant beta diversities”?

Fig 1. N=7 bold individuals?

L275. Delete first “bold”.

L301. “by personality” -> “between bold and shy”.

L309. “project” -> “study”

L312-313. “there would be significant beta diversities between them” -> beta diversities would significantly differ between them”

L313. Delete second “predicted”

L319-320. Delete “or show significant beta diversity” and exchange to “but did not differ significantly in alpha or beta diversity.” Delete “group” and “also”.

L324. Exchange to “the resulting heightened levels of cortisol can lead to…”

L326-327. Higher corticosterone than who? Also, please distinguish sentence on previous study and new finding from this study as otherwise it is confusing.

L343. Male and female juveniles, or males and females? Please clarify why.

L356-357. Latin name of species should be in italic.

L377-378. Did you mean “were more abundant in oral and skin swabs of bold turtles” ?

L388. Add a comma after “behavior”

L390. Delete “for this taxon”

L407. Delete “predicted”

L410. Add “in” before “current study”

L412. “study” -> “investigation”

L431. Do you mean “cloacal swabs” instead of “gut microbiome swabs”?

6. PLOS authors have the option to publish the peer review history of their article (what does this mean?). If published, this will include your full peer review and any attached files.

Reviewer #1: **Yes: **Jason G. Randall

Reviewer #2: No

---

## [Author Response · Author response to Decision Letter 1]

25 Nov 2025

Reviewer 1:

The references section does not include numbers that would correspond to the in-text citations. In addition, page numbers are missing.

● Numbers have been added to all references that correspond to the in-text citations. Pages have been numbered.

It is stated that a total of 12 turtles were collected but only 11 total turtles deemed bold/shy (first mentioned in Figure 1) with no mention of the 12th. It could be confused that this was either the turtle that supposedly did not emerge after 600 seconds where the trail was terminated (is it not mentioned if this turtle was included with the list of shy individuals or not), or the immature unknown sex individual for sex comparison studies, although only the 10 adults had their sexes determined; so this could be more clearly stated. In addition, cloacal, oral, and skin samples were not taken from all 12 turtles (only either 10 or 11) with no apparent mention for this reason, even though it is mentioned that samples were taken from each encountered individual.:

● Thank you for your feedback. We came to the realization that one turtle was counted twice for a portion of the analyses. We corrected our sample size to 11, re-ran all analyses, and updated the statistics presented in the results section. There were no significant statistical changes after correcting for this. At least one type of sample was taken from each individual, however we were not able to collect all three types of samples from all individuals due to post-collection failures (L277–279). No turtle was entirely excluded from this study.

In regards to Figure 2: Is there a reason only the Bray-Curtis distance analysis is shown in a figure and not the other tests for beta diversity? Additionally, this figure may work better if “Skin”, “Oral”, and “Cloacal” were shown by their respective A-C designated letters as was included in Figure 1. The text also includes “combined” as a sample type for this test which does not appear in the figure. Lastly, the ellipses are not “shaded”.:

● We chose not to include the other figures as to not overwhelm the manuscript with unnecessary graphics, but have included these in the Supplementary Information. The ellipses are now shaded. Figure 2 has been revised to show sample types by their A-C designated letters.

Although not required, consider en dashes for ranges, and not starting a sentence directly with a bracketed citation:

● En dashes are now used for ranges, and no sentences begin directly with a bracketed citation.

Reviewer 2:

Labelling throughout the manuscript should be improved. As the Authors did not conduct repeated testing (L181), labelling behaviours as ‘personality’, ‘boldness’ or ‘shyness’ is misleading, instead of using ‘personality’ as a term, the Authors should use ‘behaviour’. Furthermore, why do Authors make a distinction between “microbiome” and “microbiota” (L76-78) and then use them interchangeably in the text?

● L220–225: We have placed greater emphasis on the fact that several previous papers establish that the behavioral assay used is highly consistent in these turtles, making this behavioral measurement a reasonable proxy for personality, and added additional papers that indeed use this behavioral assay in this citation. In addition, there were logistical constraints due to concurrent research activities; we have added this statement to the manuscript as well.

● We have reviewed our use of the terms ‘microbiome’ and ‘microbiota’ and corrected the term used based on our given definition, changing some previous instances of ‘microbiome’ to ‘microbiota’.

Was the overall aim of the manuscript just to get more data on Eastern Box Turtles because they are “reptiles and long-lived, non-model organisms” (L28) or that “and that they were used because of “the dire need for more data on their behaviour and microbiome as reptiles and non-model organism” (L15-151). This is a bit weak reason.

● L32–33, 37–40 153-166: Thank you for your feedback. We have revised the text in the abstract to clarify our overall aim.

● L153-166: We have also revised the description of our primary objectives in the last paragraph of the Introduction.

Methods and data collection are unclear. For instance, it seems that some turtles were already captured and tracked beforehand (“many turtles encountered in this study had been previously captured and tracked” L157-158; “some turtles” L164), but how many turtles were captured of the ones tested in this manuscript is not precisely given. Further, how were the tests done, was the “time-to-head-emergence” test conducted first, then the measurements, then the swabs or did you follow other order of data collection? How were the turtles located? What motivated your decision to label turtles “bold” if they emerged in less than 10 and “shy” in more than 10 seconds – was there any biological reason for this? Why did the Authors conduct only one behavioural test, when they knew that repeated testing is needed to assess personality?

● L200-204: The order and rationale for the order has been specified. Head emergence assay was done first, then swab collection, then measurements. This prevented handling from influencing behavior measurements, and prevented measuring tools from contaminating the turtle prior to swabbing.

● L196–200: Turtles used in this study were not being tracked at the time of this study. Seven had been previously captured, and two of those had been previously radiotracked. This has been clarified in the manuscript.

● L220-221: We defined ‘bold’ and ‘shy’ based on a 10-second head-emergence cutoff using data from Kashon & Carlson 2018, who found mean emergence to be strongly bimodal in the same study population. Specifically, they found similar numbers of individual turtles to emerge within 7 seconds or less on average (across repeated trials) or after more than 58 seconds on average; they found no intermediate behavioral phenotypes.

● L221-228: Repeated testing: see previous justification related to the high repeatability. In addition, there were logistical constraints due to concurrent research activities. We have added this information into the manuscript

Statistical analyses should be carefully conducted and results discussed with taking a small sample size that the Authors collected in mind (only 12 turtles, uneven number of females and males, comparisons with differing numbers e.g. 7 versus 2 animals, etc).

● L63–64, 518–520: Thank you for your feedback. In our Abstract, we now include that our results should be considered in the context of limited sample sizes (L63–64). We also added a sentence that addresses this in the Conclusion (L518–520). Throughout the Discussion, we also mention that an increase in sample size would be beneficial to increase support for our results.

Statistical details should not be reported under Figure and Table descriptions, but instead in the main text of the manuscript.

● Thank you for your feedback. We found no requirements to exclude statistical details from figures and tables descriptions in the formatting requirements from PLOS One. We choose to have those details in the captions to help the figures and tables stand alone from the main text.

The analyses conducted with linking behaviour and microbiome are not very clearly described, so the reader will be left somewhat puzzled what was done where. Do these analyses refer to L210-212? Which analyses were done to check the effect of personality, sex, age etc on microbiome? L250. What kind of interaction was found in terms of personality type & sex? L333-334. Here it seems that sex effects are convoluted with age, as IDs of unknown sex were also the juvenile and subadult ones? Please clarify.

● L302–311, 382–424: Thank you for your feedback. We edited the text (L302–311) to indicate in more detail the specific test and result found between the microbiome to sex and/or personality. In the Discussion (L382–424) we included more detail on the relationship between the microbiome, sex, and age.

Literature coverage was made somewhat superficially, without giving specific examples. This is particularly true for Abstract (very generally written without discussion and conclusion sentences, giving general claims on phenomena that are quite underexplored; e.g. L21-22, L23-24, L24-25, L44-45). In general, details of previous studies are missing from the manuscript.

● Thank you for your feedback. We heavily edited the Abstract to better summarize the content of the paper and meet the 300 word-limit requirement. We now have a Discussion sentence that notes our limited sample size and better contextualizes our results. In our Introduction, we go into more detail on the general phenomena that seem underexplored in our limited Abstract (i.e., individual personality shaping an ecosystem [L77-80], boldness/shyness [L84-94], microbial composition [95-105], the influence of the gut microbiome [L106-116] and specific examples [L117-128]).

Related point is that it is not clear from the Discussion what the Results represent and why they are important.

● L373-374, 514-521: Thank you for your feedback. Additional information has been added to the beginning of the discussion (L373-374) and the conclusion (L514–521) to help convey the results, and their implications and importance.

Conclusion and overall messages that the reader can get from the manuscript are not clearly conveyed.

● Thank you for your feedback. Additional information has been added to the conclusion (L514–521) to help convey the results, their implications and importance, and limitations.

References are very messy and unnumbered which makes it difficult to follow numbering in text. There is no consistent style in capitalization of words within a title. Editor names are given for journal articles which is not needed, links that are not doi identifiers, species names are not in italic, journal names are not capitalized etc.

• Capitalization has been fixed to sentence-case. Editors are now only named in cited chapters in books, as per ICJME reference examples on the NIH website (https://www.nlm.nih.gov/bsd/uniform_requirements.html#journals). Links that are not DOI identifiers are now cited for online journal articles only, as per ICJME references on the NIH website (see above link). All species names are now in italics, and journal names are now in NLM abbreviated form and capitalized.

L50-51 is redundant with L61-64. – Corrected; lines 61-64 have been omitted.

L54. “to explain variation in personality” can be omitted. – Corrected

L135. What does it mean that “the microbiomes will have significant beta diversities”? – Corrected; further context added (L160–162)

Fig 1. N=7 bold individuals? – Corrected to N = 6

L275. Delete first “bold”. – Corrected

L301. “by personality” -> “between bold and shy”. – Corrected

L309. “project” -> “study” – Corrected

L312-313. “there would be significant beta diversities between them” -> beta diversities would significantly differ between them” – Corrected

L313. Delete second “predicted” – Corrected

L319-320. Delete “or show significant beta diversity” and exchange to “but did not differ significantly in alpha or beta diversity.” Delete “group” and “also”. – Corrected

L324. Exchange to “the resulting heightened levels of cortisol can lead to…” – Corrected

L326-327. Higher corticosterone than who? Also, please distinguish sentence on previous study and new finding from this study as otherwise it is confusing. – Corrected; clarified (L387-396)

L343. Male and female juveniles, or males and females? Please clarify why. – Corrected; clarified (L 408–412).

L356-357. Latin name of species should be in italic. – Corrected

L377-378. Did you mean “were more abundant in oral and skin swabs of bold turtles” ? – Yes; corrected

L388. Add a comma after “behavior” – Corrected

L390. Delete “for this taxon” – Corrected

L407. Delete “predicted” – Corrected

L410. Add “in” before “current study” – Corrected

L412. “study” -> “investigation” – Corrected

L431. Do you mean “cloacal swabs” instead of “gut microbiome swabs”? – Yes; corrected

---

## [Editor Report · Decision Letter 1]

2 Dec 2025

The Relationships Between Box Turtle Gut Microbiomes and Personality

PONE-D-25-40604R1

Dear Dr. Harlow,

We’re pleased to inform you that your manuscript has been judged scientifically suitable for publication and will be formally accepted for publication once it meets all outstanding technical requirements.

Kind regards,

Bernd Schierwater, Ph.D

Academic Editor

PLOS ONE
---

## [Editor Report · Acceptance letter]

PONE-D-25-40604R1

PLOS One

Dear Dr. Harlow,

I'm pleased to inform you that your manuscript has been deemed suitable for publication in PLOS One. Congratulations! Your manuscript is now being handed over to our production team.

Kind regards,

on behalf of

Prof. Bernd Schierwater

Academic Editor

PLOS One